# Bayesian Compression for Deep Learning

**Christos Louizos**
University of Amsterdam
TNO Intelligent Imaging
`c.louizos@uva.nl`

**Karen Ullrich**
University of Amsterdam
`k.ullrich@uva.nl`

**Max Welling**
University of Amsterdam
CIFAR*
`m.welling@uva.nl`

## Abstract

Compression and computational efficiency in deep learning have become a problem of great significance. In this work, we argue that the most principled and effective way to attack this problem is by adopting a Bayesian point of view, where through sparsity inducing priors we prune large parts of the network. We introduce two novelties in this paper: 1) we use hierarchical priors to prune nodes instead of individual weights, and 2) we use the posterior uncertainties to determine the optimal fixed point precision to encode the weights. Both factors significantly contribute to achieving the state of the art in terms of compression rates, while still staying competitive with methods designed to optimize for speed or energy efficiency.

## 1 Introduction

While deep neural networks have become extremely successful in in a wide range of applications, often exceeding human performance, they remain difficult to apply in many real world scenarios. For instance, making billions of predictions per day comes with substantial energy costs given the energy consumption of common Graphical Processing Units (GPUs). Also, real-time predictions are often about a factor 100 away in terms of speed from what deep NNs can deliver, and sending NNs with millions of parameters through band limited channels is still impractical. As a result, running them on hardware limited devices such as smart phones, robots or cars requires substantial improvements on all of these issues. For all those reasons, compression and efficiency have become a topic of interest in the deep learning community.

While all of these issues are certainly related, compression and performance optimizing procedures might not always be aligned. As an illustration, consider the convolutional layers of Alexnet, which account for only 4% of the parameters but 91% of the computation [65]. Compressing these layers will not contribute much to the overall memory footprint.

There is a variety of approaches to address these problem settings. However, most methods have the common strategy of reducing both the neural network structure and the effective fixed point precision for each weight. A justification for the former is the finding that NNs suffer from significant parameter redundancy [14]. Methods in this line of thought are network pruning, where unnecessary connections are being removed [38, 24, 21], or student-teacher learning where a large network is used to train a significantly smaller network [5, 26].

From a Bayesian perspective network pruning and reducing bit precision for the weights is aligned with achieving high accuracy, because Bayesian methods search for the optimal model structure (which leads to pruning with sparsity inducing priors), and reward uncertain posteriors over parameters through the bits back argument [27] (which leads to removing insignificant bits). This relation is made explicit in the MDL principle [20] which is known to be related to Bayesian inference.

In this paper we will use the variational Bayesian approximation for Bayesian inference which has also been explicitly interpreted in terms of model compression [27]. By employing sparsity inducing priors for hidden units (and not individual weights) we can prune neurons including all their ingoing and outgoing weights. This avoids more complicated and inefficient coding schemes needed for pruning or vector quantizing individual weights. As an additional Bayesian bonus we can use the variational posterior uncertainty to assess which bits are significant and remove the ones which fluctuate too much under approximate posterior sampling. From this we derive the optimal fixed point precision per layer, which is still practical on chip.

## 2    Variational Bayes and Minimum Description Length

A fundamental theorem in information theory is the minimum description length (MDL) principle [20]. It relates to compression directly in that it defines the best hypothesis to be the one that communicates the sum of the model (complexity cost $\mathcal{L}^C$) and the data misfit (error cost $\mathcal{L}^E$) with the minimum number of bits [57, 58]. It is well understood that variational inference can be reinterpreted from an MDL point of view [54, 69, 27, 29, 19]. More specifically, assume that we are presented with a dataset $\mathcal{D}$ that consists from $N$ input-output pairs $\{(\mathbf{x}_1, y_1), \ldots, (\mathbf{x}_n, y_n)\}$. Let $p(\mathcal{D}|\mathbf{w}) = \prod_{i=1}^N p(y_i|\mathbf{x}_i, \mathbf{w})$ be a parametric model, e.g. a deep neural network, that maps inputs $\mathbf{x}$ to their corresponding outputs $y$ using parameters $\mathbf{w}$ governed by a prior distribution $p(\mathbf{w})$. In this scenario, we wish to approximate the intractable posterior distribution $p(\mathbf{w}|\mathcal{D}) = p(\mathcal{D}|\mathbf{w})p(\mathbf{w})/p(\mathcal{D})$ with a fixed form approximate posterior $q_\phi(\mathbf{w})$ by optimizing the variational parameters $\phi$ according to:

$$\mathcal{L}(\phi) = \underbrace{\mathbb{E}_{q_\phi(\mathbf{w})}[\log p(\mathcal{D}|\mathbf{w})]}_{\mathcal{L}^E} + \underbrace{\mathbb{E}_{q_\phi(\mathbf{w})}[\log p(\mathbf{w})] + \mathcal{H}(q_\phi(\mathbf{w}))}_{\mathcal{L}^C}, \tag{1}$$

where $\mathcal{H}(\cdot)$ denotes the entropy and $\mathcal{L}(\phi)$ is known as the evidence-lower-bound (ELBO) or negative variational free energy. As indicated in eq. 1, $\mathcal{L}(\phi)$ naturally decomposes into a minimum cost for communicating the targets $\{y_n\}_{n=1}^N$ under the assumption that the sender and receiver agreed on a prior $p(\mathbf{w})$ and that the receiver knows the inputs $\{\mathbf{x}_n\}_{n=1}^N$ and form of the parametric model.

By using sparsity inducing priors for groups of weights that feed into a neuron the Bayesian mechanism will start pruning hidden units that are not strictly necessary for prediction and thus achieving compression. But there is also a second mechanism by which Bayes can help us compress. By explicitly entertaining noisy weight encodings through $q_\phi(\mathbf{w})$ we can benefit from the bits-back argument [27, 29] due to the entropy term; this is in contrast to infinitely precise weights that lead to $\mathcal{H}(\delta(\mathbf{w})) = -\infty^2$. Nevertheless in practice, the data misfit term $\mathcal{L}^E$ is intractable for neural network models under a noisy weight encoding, so as a solution Monte Carlo integration is usually employed. Continuous $q_\phi(\mathbf{w})$ allow for the reparametrization trick [34, 56]. Here, we replace sampling from $q_\phi(\mathbf{w})$ by a deterministic function of the variational parameters $\phi$ and random samples from some noise variables $\epsilon$:

$$\mathcal{L}(\phi) = \mathbb{E}_{p(\epsilon)}[\log p(\mathcal{D}|f(\phi, \epsilon))] + \mathbb{E}_{q_\phi(\mathbf{w})}[\log p(\mathbf{w})] + \mathcal{H}(q_\phi(\mathbf{w})), \tag{2}$$

where $\mathbf{w} = f(\phi, \epsilon)$. By applying this trick, we obtain unbiased stochastic gradients of the ELBO with respect to the variational parameters $\phi$, thus resulting in a standard optimization problem that is fit for stochastic gradient ascent. The efficiency of the gradient estimator resulting from eq. 2 can be further improved for neural networks by utilizing local reparametrizations [35] (which we will use in our experiments); they provide variance reduction in an efficient way by locally marginalizing the weights at each layer and instead sampling the distribution of the pre-activations.

## 3    Related Work

One of the earliest ideas and most direct approaches to tackle efficiency is pruning. Originally introduced by [38], pruning has recently been demonstrated to be applicable to modern architectures [25, 21]. It had been demonstrated that an overwhelming amount of up to 99,5% of parameters can be pruned in common architectures. There have been quite a few encouraging results obtained by (empirical) Bayesian approaches that employ weight pruning [19, 7, 50, 67, 49]. Nevertheless,

weight pruning is in general inefficient for compression since the matrix format of the weights is not taken into consideration, therefore the Compressed Sparse Column (CSC) format has to be employed. Moreover, note that in conventional CNNs most flops are used by the convolution operation. Inspired by this observation, several authors proposed pruning schemes that take these considerations into account [70, 71] or even go as far as efficiency aware architectures to begin with [31, 15, 30]. From the Bayesian viewpoint, similar pruning schemes have been explored at [45, 51, 37, 33].

Given optimal architecture, NNs can further be compressed by quantization. More precisely, there are two common techniques. First, the set of accessible weights can be reduced drastically. As an extreme example, [13, 46, 55, 72] and [11] trained NN to use only binary or tertiary weights with floating point gradients. This approach however is in need of significantly more parameters than their ordinary counterparts. Work by [18] explores various techniques beyond binary quantization: k-means quantization, product quantization and residual quantization. Later studies extent this set to optimal fixed point [42] and hashing quantization [10]. [25] apply k-means clustering and consequent center training. From a practical point of view, however, all these are fairly unpractical during test time. For the computation of each feature map in a net, the original weight matrix must be reconstructed from the indexes in the matrix and a codebook that contains all the original weights. This is an expensive operation and this is why some studies propose a different approach than set quantization. Precision quantization simply reduces the bit size per weight. This has a great advantage over set quantization at inference time since feature maps can simply be computed with less precision weights. Several studies show that this has little to no effect on network accuracy when using 16bit weights [47, 22, 12, 68, 9]. Somewhat orthogonal to the above discussion but certainly relevant are approaches that customize the implementation of CNNs for hardware limited devices[30, 4, 60].

## 4   Bayesian compression with scale mixtures of normals

Consider the following prior over a parameter $w$ where its scale $z$ is governed by a distribution $p(z)$:

$$z \sim p(z); \qquad w \sim \mathcal{N}(w; 0, z^2), \tag{3}$$

with $z^2$ serving as the variance of the zero-mean normal distribution over $w$. By treating the scales of $w$ as random variables we can recover marginal prior distributions over the parameters that have heavier tails and more mass at zero; this subsequently biases the posterior distribution over $w$ to be sparse. This family of distributions is known as scale-mixtures of normals [6, 2] and it is quite general, as a lot of well known sparsity inducing distributions are special cases.

One example of the aforementioned framework is the spike-and-slab distribution [48], the golden standard for sparse Bayesian inference. Under the spike-and-slab, the mixing density of the scales is a Bernoulli distribution, thus the marginal $p(w)$ has a delta "spike" at zero and a continuous "slab" over the real line. Unfortunately, this prior leads to a computationally expensive inference since we have to explore a space of $2^M$ models, where $M$ is the number of the model parameters. Dropout [28, 64], one of the most popular regularization techniques for neural networks, can be interpreted as positing a spike and slab distribution over the weights where the variance of the "slab" is zero [17, 43]. Another example is the Laplace distribution which arises by considering $p(z^2) = \text{Exp}(\lambda)$. The mode of the posterior distribution under a Laplace prior is known as the Lasso [66] estimator and has been previously used for sparsifying neural networks at [70, 59]. While computationally simple, the Lasso estimator is prone to "shrinking" large signals [8] and only provides point estimates about the parameters. As a result it does not provide uncertainty estimates, it can potentially overfit and, according to the bits-back argument, is inefficient for compression.

For these reasons, in this paper we will tackle the problem of compression and efficiency in neural networks by adopting a Bayesian treatment and inferring an approximate posterior distribution over the parameters under a scale mixture prior. We will consider two choices for the prior over the scales $p(z)$; the hyperparameter free log-uniform prior [16, 35] and the half-Cauchy prior, which results into a horseshoe [8] distribution. Both of these distributions correspond to a continuous relaxation of the spike-and-slab prior and we provide a brief discussion on their shrinkage properties at Appendix C.

## 4.1 Reparametrizing variational dropout for group sparsity

One potential choice for $p(z)$ is the improper log-uniform prior [35]: $p(z) \propto |z|^{-1}$. It turns out that we can recover the log-uniform prior over the weights $w$ if we marginalize over the scales $z$:

$$p(w) \propto \int \frac{1}{|z|} \mathcal{N}(w|0, z^2) dz = \frac{1}{|w|}. \tag{4}$$

This alternative parametrization of the log uniform prior is known in the statistics literature as the normal-Jeffreys prior and has been introduced by [16]. This formulation allows to "couple" the scales of weights that belong to the same group (e.g. neuron or feature map), by simply sharing the corresponding scale variable $z$ in the joint prior[3]:

$$p(\mathbf{W}, \mathbf{z}) \propto \prod_i^A \frac{1}{|z_i|} \prod_{ij}^{A,B} \mathcal{N}(w_{ij}|0, z_i^2), \tag{5}$$

where $\mathbf{W}$ is the weight matrix of a fully connected neural network layer with $A$ being the dimensionality of the input and $B$ the dimensionality of the output. Now consider performing variational inference with a joint approximate posterior parametrized as follows:

$$q_\phi(\mathbf{W}, \mathbf{z}) = \prod_{i=1}^A \mathcal{N}(z_i|\mu_{z_i}, \mu_{z_i}^2 \alpha_i) \prod_{i,j}^{A,B} \mathcal{N}(w_{ij}|z_i\mu_{ij}, z_i^2 \sigma_{ij}^2), \tag{6}$$

where $\alpha_i$ is the dropout rate [64, 35, 49] of the given group. As explained at [35, 49], the multiplicative parametrization of the approximate posterior over $\mathbf{z}$ suffers from high variance gradients; therefore we will follow [49] and re-parametrize it in terms of $\sigma_{z_i}^2 = \mu_{z_i}^2 \alpha_i$, hence optimize w.r.t. $\sigma_{z_i}^2$. The lower bound under this prior and approximate posterior becomes:

$$\mathcal{L}(\phi) = \mathbb{E}_{q_\phi(\mathbf{z})q_\phi(\mathbf{W}|\mathbf{z})}[\log p(\mathcal{D}|\mathbf{W})] - \mathbb{E}_{q_\phi(\mathbf{z})}[KL(q_\phi(\mathbf{W}|\mathbf{z})||p(\mathbf{W}|\mathbf{z}))] - KL(q_\phi(\mathbf{z})||p(\mathbf{z})). \tag{7}$$

Under this particular variational posterior parametrization the negative KL-divergence from the conditional prior $p(\mathbf{W}|\mathbf{z})$ to the approximate posterior $q_\phi(\mathbf{W}|\mathbf{z})$ is independent of $\mathbf{z}$:

$$KL(q_\phi(\mathbf{W}|\mathbf{z})||p(\mathbf{W}|\mathbf{z})) = \frac{1}{2} \sum_{i,j}^{A,B} \left( \log \frac{\cancel{z_i^2}}{\cancel{z_i^2} \sigma_{ij}^2} + \frac{\cancel{z_i^2} \sigma_{ij}^2}{\cancel{z_i^2}} + \frac{\cancel{z_i^2} \mu_{ij}^2}{\cancel{z_i^2}} - 1 \right). \tag{8}$$

This independence can be better understood if we consider a non-centered parametrization of the prior [53]. More specifically, consider reparametrizing the weights as $\tilde{w}_{ij} = \frac{w_{ij}}{z_i}$; this will then result into $p(\mathbf{W}|\mathbf{z})p(\mathbf{z}) = p(\tilde{\mathbf{W}})p(\mathbf{z})$, where $p(\tilde{\mathbf{W}}) = \prod_{i,j} \mathcal{N}(\tilde{w}_{ij}|0, 1)$ and $\mathbf{W} = \text{diag}(\mathbf{z})\tilde{\mathbf{W}}$. Now if we perform variational inference under the $p(\tilde{\mathbf{W}})p(\mathbf{z})$ prior with an approximate posterior that has the form of $q_\phi(\tilde{\mathbf{W}}, \mathbf{z}) = q_\phi(\tilde{\mathbf{W}})q_\phi(\mathbf{z})$, with $q_\phi(\tilde{\mathbf{W}}) = \prod_{i,j} \mathcal{N}(\tilde{w}_{ij}|\mu_{ij}, \sigma_{ij}^2)$, then we see that we arrive at the same expressions for the negative KL-divergence from the prior to the approximate posterior. Finally, the negative KL-divergence from the normal-Jeffreys scale prior $p(\mathbf{z})$ to the Gaussian variational posterior $q_\phi(\mathbf{z})$ depends only on the "implied" dropout rate, $\alpha_i = \sigma_{z_i}^2 / \mu_{z_i}^2$, and takes the following form [49]:

$$- KL(q_\phi(\mathbf{z})||p(\mathbf{z})) \approx \sum_i^A \left( k_1 \sigma(k_2 + k_3 \log \alpha_i) - 0.5m(-\log \alpha_i) - k_1 \right), \tag{9}$$

where $\sigma(\cdot)$, $m(\cdot)$ are the sigmoid and softplus functions respectively[4] and $k_1 = 0.63576$, $k_2 = 1.87320$, $k_3 = 1.48695$. We can now prune entire groups of parameters by simply specifying a threshold for the variational dropout rate of the corresponding group, e.g. $\log \alpha_i = (\log \sigma_{z_i}^2 - \log \mu_{z_i}^2) \geq t$. It should be mentioned that this prior parametrization readily allows for a more flexible marginal posterior over the weights as we now have a compound distribution, $q_\phi(\mathbf{W}) = \int q_\phi(\mathbf{W}|\mathbf{z})q_\phi(\mathbf{z})d\mathbf{z}$; this is in contrast to the original parametrization and the Gaussian approximations employed by [35, 49].

Furthermore, this approach generalizes the low variance additive parametrization of variational dropout proposed for weight sparsity at [49] to group sparsity (which was left as an open question at [49]) in a principled way.

At test time, in order to have a single feedforward pass we replace the distribution over $\mathbf{W}$ at each layer with a single weight matrix, the masked variational posterior mean:

$$\hat{\mathbf{W}} = \text{diag}(\mathbf{m})\, \mathbb{E}_{q(\mathbf{z})q(\tilde{\mathbf{W}})}[\text{diag}(\mathbf{z})\tilde{\mathbf{W}}] = \text{diag}\left(\mathbf{m} \odot \boldsymbol{\mu}_z\right)\mathbf{M}_W, \tag{10}$$

where $\mathbf{m}$ is a binary mask determined according to the group variational dropout rate and $\mathbf{M}_W$ are the means of $q_\phi(\tilde{\mathbf{W}})$. We further use the variational posterior marginal variances[5] for this particular posterior approximation:

$$\mathbb{V}(w_{ij})_{NJ} = \sigma_{z_i}^2\left(\sigma_{ij}^2 + \mu_{ij}^2\right) + \sigma_{ij}^2\mu_{z_i}^2, \tag{11}$$

to asess the bit precision of each weight in the weight matrix. More specifically, we employed the mean variance across the weight matrix $\hat{\mathbf{W}}$ to compute the unit round off necessary to represent the weights. This method will give us the amount significant bits, and by adding 3 exponent and 1 sign bits we arrive at the final bit precision for the entire weight matrix $\hat{\mathbf{W}}$[6]. We provide more details at Appendix B.

## 4.2 Group horseshoe with half-Cauchy scale priors

Another choice for $p(z)$ is a proper half-Cauchy distribution: $\mathcal{C}^+(0,s) = 2(s\pi(1 + (z/s)^2))^{-1}$; it induces a horseshoe prior [8] distribution over the weights, which is a well known sparsity inducing prior in the statistics literature. More formally, the prior hierarchy over the weights is expressed as (in a non-centered parametrization):

$$s \sim \mathcal{C}^+(0,\tau_0); \qquad \tilde{z}_i \sim \mathcal{C}^+(0,1); \qquad \tilde{w}_{ij} \sim \mathcal{N}(0,1); \qquad w_{ij} = \tilde{w}_{ij}\tilde{z}_i s, \tag{12}$$

where $\tau_0$ is the free parameter that can be tuned for specific desiderata. The idea behind the horseshoe is that of the "global-local" shrinkage; the global scale variable $s$ pulls all of the variables towards zero whereas the heavy tailed local variables $z_i$ can compensate and allow for some weights to escape. Instead of directly working with the half-Cauchy priors we will employ a decomposition of the half-Cauchy that relies upon (inverse) gamma distributions [52] as this will allow us to compute the negative KL-divergence from the scale prior $p(\mathbf{z})$ to an approximate log-normal scale posterior $q_\phi(\mathbf{z})$ in closed form (the derivation is given in Appendix D). More specifically, we have that the half-Cauchy prior can be expressed in a non-centered parametrization as:

$$p(\tilde{\beta}) = \mathcal{IG}(0.5,1); \qquad p(\tilde{\alpha}) = \mathcal{G}(0.5,k^2); \qquad z^2 = \tilde{\alpha}\tilde{\beta}, \tag{13}$$

where $\mathcal{IG}(\cdot,\cdot), \mathcal{G}(\cdot,\cdot)$ correspond to the inverse Gamma and Gamma distributions in the scale parametrization, and $z$ follows a half-Cauchy distribution with scale $k$. Therefore we will re-express the whole hierarchy as:

$$s_b \sim \mathcal{IG}(0.5,1); \quad s_a \sim \mathcal{G}(0.5,\tau_0^2); \quad \tilde{\beta}_i \sim \mathcal{IG}(0.5,1); \quad \tilde{\alpha}_i \sim \mathcal{G}(0.5,1); \quad \tilde{w}_{ij} \sim \mathcal{N}(0,1);$$

$$w_{ij} = \tilde{w}_{ij}\sqrt{s_a s_b \tilde{\alpha}_i \tilde{\beta}_i}. \tag{14}$$

It should be mentioned that the improper log-uniform prior is the limiting case of the horseshoe prior when the shapes of the (inverse) Gamma hyperpriors on $\tilde{\alpha}_i, \tilde{\beta}_i$ go to zero [8]. In fact, several well known shrinkage priors can be expressed in this form by altering the shapes of the (inverse) Gamma hyperpriors [3]. For the variational posterior we will employ the following mean field approximation:

$$q_\phi(s_b, s_a, \tilde{\boldsymbol{\beta}}) = \mathcal{LN}(s_b|\mu_{s_b},\sigma_{s_b}^2)\mathcal{LN}(s_a|\mu_{s_a},\sigma_{s_a}^2)\prod_i^A \mathcal{LN}(\tilde{\beta}_i|\mu_{\tilde{\beta}_i},\sigma_{\tilde{\beta}_i}^2) \tag{15}$$

$$q_\phi(\tilde{\boldsymbol{\alpha}}, \tilde{\mathbf{W}}) = \prod_i^A \mathcal{LN}(\tilde{\alpha}_i|\mu_{\tilde{\alpha}_i},\sigma_{\tilde{\alpha}_i}^2)\prod_{i,j}^{A,B}\mathcal{N}(\tilde{w}_{ij}|\mu_{\tilde{w}_{ij}},\sigma_{\tilde{w}_{ij}}^2), \tag{16}$$

where $\mathcal{LN}(\cdot, \cdot)$ is a log-normal distribution. It should be mentioned that a similar form of non-centered variational inference for the horseshoe has been also successfully employed for undirected models at [32]. Notice that we can also apply local reparametrizations [35] when we are sampling $\sqrt{\tilde{\alpha}_i \tilde{\beta}_i}$ and $\sqrt{s_a s_b}$ by exploiting properties of the log-normal distribution[7] and thus forming the implied:

$$\tilde{z}_i = \sqrt{\tilde{\alpha}_i \tilde{\beta}_i} \sim \mathcal{LN}(\mu_{\tilde{z}_i}, \sigma_{\tilde{z}_i}^2); \qquad s = \sqrt{s_a s_b} \sim \mathcal{LN}(\mu_s, \sigma_s^2) \qquad (17)$$

$$\mu_{\tilde{z}_i} = \frac{1}{2}(\mu_{\tilde{\alpha}_i} + \mu_{\tilde{\beta}_i}); \quad \sigma_{\tilde{z}_i}^2 = \frac{1}{4}(\sigma_{\tilde{\alpha}_i}^2 + \sigma_{\tilde{\beta}_i}^2); \quad \mu_s = \frac{1}{2}(\mu_{s_a} + \mu_{s_b}); \quad \sigma_s^2 = \frac{1}{4}(\sigma_{s_a}^2 + \sigma_{s_b}^2). \quad (18)$$

As a threshold rule for group pruning we will use the negative log-mode[8] of the local log-normal r.v. $z_i = s\tilde{z}_i$, i.e. prune when $(\sigma_{z_i}^2 - \mu_{z_i}) \geq t$, with $\mu_{z_i} = \mu_{\tilde{z}_i} + \mu_s$ and $\sigma_{z_i}^2 = \sigma_{\tilde{z}_i}^2 + \sigma_s^2$. This ignores dependencies among the $z_i$ elements induced by the common scale $s$, but nonetheless we found that it works well in practice. Similarly with the group normal-Jeffreys prior, we will replace the distribution over $\mathbf{W}$ at each layer with the masked variational posterior mean during test time:

$$\hat{\mathbf{W}} = \text{diag}(\mathbf{m}) \, \mathbb{E}_{q(\mathbf{z})q(\tilde{\mathbf{W}})}[\text{diag}(\mathbf{z})\tilde{\mathbf{W}}] = \text{diag}\left(\mathbf{m} \odot \exp(\boldsymbol{\mu}_z + \frac{1}{2}\boldsymbol{\sigma}_z^2)\right)\mathbf{M}_W, \qquad (19)$$

where $\mathbf{m}$ is a binary mask determined according to the aforementioned threshold, $\mathbf{M}_W$ are the means of $q(\tilde{\mathbf{W}})$ and $\boldsymbol{\mu}_z, \boldsymbol{\sigma}_z^2$ are the means and variances of the local log-normals over $z_i$. Furthermore, similarly to the group normal-Jeffreys approach, we will use the variational posterior marginal variances:

$$\mathbb{V}(w_{ij})_{HS} = (\exp(\sigma_{z_i}^2) - 1)\exp(2\mu_{z_i} + \sigma_{z_i}^2)(\sigma_{ij}^2 + \mu_{ij}^2) + \sigma_{ij}^2 \exp(2\mu_{z_i} + \sigma_{z_i}^2), \quad (20)$$

to compute the final bit precision for the entire weight matrix $\hat{\mathbf{W}}$.

## 5 Experiments

We validated the compression and speed-up capabilities of our models on the well-known architectures of LeNet-300-100 [39], LeNet-5-Caffe[9] on MNIST [40] and, similarly with [49], VGG [61][10] on CIFAR 10 [36]. The groups of parameters were constructed by coupling the scale variables for each filter for the convolutional layers and for each input neuron for the fully connected layers. We provide the algorithms that describe the forward pass using local reparametrizations for fully connected and convolutional layers with each of the employed approximate posteriors at appendix F. For the horseshoe prior we set the scale $\tau_0$ of the global half-Cauchy prior to a reasonably small value, e.g. $\tau_0 = 1e - 5$. This further increases the prior mass at zero, which is essential for sparse estimation and compression. We also found that constraining the standard deviations as described at [44] and "warm-up" [62] helps in avoiding bad local optima of the variational objective. Further details about the experimental setup can be found at Appendix A. Determining the threshold for pruning can be easily done with manual inspection as usually there are two well separated clusters (signal and noise). We provide a sample visualization at Appendix E.

### 5.1 Architecture learning & bit precisions

We will first demonstrate the group sparsity capabilities of our methods by illustrating the learned architectures at Table 1, along with the inferred bit precision per layer. As we can observe, our methods infer significantly smaller architectures for the LeNet-300-100 and LeNet-5-Caffe, compared to Sparse Variational Dropout, Generalized Dropout and Group Lasso. Interestingly, we observe that for the VGG network almost all of big 512 feature map layers are drastically reduced to around 10 feature maps whereas the initial layers are mostly kept intact. Furthermore, all of the Bayesian methods considered require far fewer than the standard 32 bits per-layer to represent the weights, sometimes even allowing for 5 bit precisions.

Table 1: Learned architectures with Sparse VD [49], Generalized Dropout (GD) [63] and Group Lasso (GL) [70]. Bayesian Compression (BC) with group normal-Jeffreys (BC-GNJ) and group horseshoe (BC-GHS) priors correspond to the proposed models. We show the amount of neurons left after pruning along with the average bit precisions for the weights at each layer.

| Network & size | Method | Pruned architecture | Bit-precision |
|---|---|---|---|
| LeNet-300-100 | Sparse VD | 512-114-72 | 8-11-14 |
| 784-300-100 | BC-GNJ | 278-98-13 | 8-9-14 |
| | BC-GHS | 311-86-14 | 13-11-10 |
| LeNet-5-Caffe | Sparse VD | 14-19-242-131 | 13-10-8-12 |
| | GD | 7-13-208-16 | - |
| 20-50-800-500 | GL | 3-12-192-500 | - |
| | BC-GNJ | 8-13-88-13 | 18-10-7-9 |
| | BC-GHS | 5-10-76-16 | 10-10–14-13 |
| VGG | BC-GNJ | 63-64-128-128-245-155-63- -26-24-20-14-12-11-11-15 | 10-10-10-10-8-8-8- -5-5-5-5-5-6-7-11 |
| $(2\times 64)$-$(2\times 128)$- -$(3\times256)$-$(8\times 512)$ | BC-GHS | 51-62-125-128-228-129-38- -13-9-6-5-6-6-6-20 | 11-12-9-14-10-8-5- -5-6-6-6-8-11-17-10 |

## 5.2 Compression Rates

For the actual compression task we compare our method to current work in three different scenarios: (i) compression achieved only by pruning, here, for non-group methods we use the CSC format to store parameters; (ii) compression based on the former but with reduced bit precision per layer (only for the weights); and (iii) the maximum compression rate as proposed by [25]. We believe

Table 2: Compression results for our methods. "DC" corresponds to Deep Compression method introduced at [25], "DNS" to the method of [21] and "SWS" to the Soft-Weight Sharing of [67]. Numbers marked with * are best case guesses.

| Model Original Error % | Method | $\frac{|\mathbf{w}\neq0|}{|\mathbf{w}|}\%$ | Compression Rates (Error %) | | |
|---|---|---|---|---|---|
| | | | Pruning | Fast Prediction | Maximum Compression |
| LeNet-300-100 | DC | 8.0 | 6 (1.6) | - | 40 (1.6) |
| | DNS | 1.8 | 28* (2.0) | - | - |
| 1.6 | SWS | 4.3 | 12* (1.9) | - | 64(1.9) |
| | Sparse VD | 2.2 | 21(1.8) | 84(1.8) | 113 (1.8) |
| | BC-GNJ | 10.8 | 9(1.8) | 36(1.8) | 58(1.8) |
| | BC-GHS | 10.6 | 9(1.8) | 23(1.9) | 59(2.0) |
| LeNet-5-Caffe | DC | 8.0 | 6*(0.7) | - | 39(0.7) |
| | DNS | 0.9 | 55*(0.9) | - | 108(0.9) |
| 0.9 | SWS | 0.5 | 100*(1.0) | - | 162(1.0) |
| | Sparse VD | 0.7 | 63(1.0) | 228(1.0) | 365(1.0) |
| | BC-GNJ | 0.9 | 108(1.0) | 361(1.0) | 573(1.0) |
| | BC-GHS | 0.6 | 156(1.0) | 419(1.0) | 771(1.0) |
| VGG | BC-GNJ | 6.7 | 14(8.6) | 56(8.8) | 95(8.6) |
| 8.4 | BC-GHS | 5.5 | 18(9.0) | 59(9.0) | 116(9.2) |

these to be relevant scenarios because (i) can be applied with already existing frameworks such as Tensorflow [1], (ii) is a practical scheme given upcoming GPUs and frameworks will be designed to work with low and mixed precision arithmetics [41, 23]. For (iii), we perform k-means clustering on the weights with k=32 and consequently store a weight index that points to a codebook of available

weights. Note that the latter achieves highest compression rate but it is however fairly unpractical at test time since the original matrix needs to be restored for each layer. As we can observe at Table 2, our methods are competitive with the state-of-the art for LeNet-300-100 while offering significantly better compression rates on the LeNet-5-Caffe architecture, without any loss in accuracy. Do note that group sparsity and weight sparsity can be combined so as to further prune some weights when a particular group is not removed, thus we can potentially further boost compression performance at e.g. LeNet-300-100. For the VGG network we observe that training from a random initialization yielded consistently less accuracy (around 1%-2% less) compared to initializing the means of the approximate posterior from a pretrained network, similarly with [49], thus we only report the latter results[11]. After initialization we trained the VGG network regularly for 200 epochs using Adam with the default hyperparameters. We observe a small drop in accuracy for the final models when using the deterministic version of the network for prediction, but nevertheless averaging across multiple samples restores the original accuracy. Note, that in general we can maintain the original accuracy on VGG without sampling by simply finetuning with a small learning rate, as done at [49]. This will still induce (less) sparsity but unfortunately it does not lead to good compression as the bit precision remains very high due to not appropriately increasing the marginal variances of the weights.

## 5.3 Speed and energy consumption

We demonstrate that our method is competitive with [70], denoted as GL, a method that explicitly prunes convolutional kernels to reduce compute time. We measure the time and energy consumption of one forward pass of a mini-batch with batch size 8192 through LeNet-5-Caffe. We average over $10^4$ forward passes and all experiments were run with Tensorflow 1.0.1, cuda 8.0 and respective cuDNN. We apply 16 CPUs run in parallel (CPU) or a Titan X (GPU). Note that we only use the pruned architecture as lower bit precision would further increase the speed-up but is not implementable in any common framework. Further, all methods we compare to in the latter experiments would barely show an improvement at all since they do not learn to prune groups but only parameters. In figure 1 we present our results. As to be expected the largest effect on the speed up is caused by GPU usage. However, both our models and best competing models reach a speed up factor of around $8\times$. We can further save about $3 \times$ energy costs by applying our architecture instead of the original one on a GPU. For larger networks the speed-up is even higher: for the VGG experiments with batch size 256 we have a speed-up factor of $51\times$.

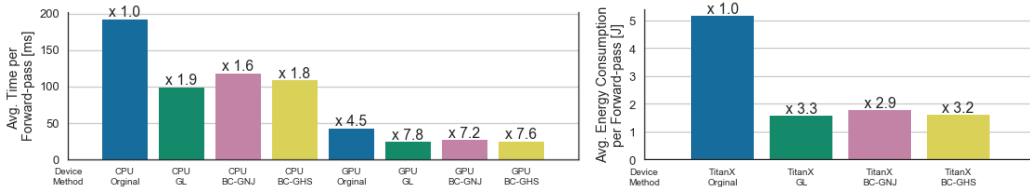

Figure 1: **Left:** Avg. Time a batch of 8192 samples takes to pass through LeNet-5-Caffe. Numbers on top of the bars represent speed-up factor relative to the CPU implementation of the original network. **Right:** Energy consumption of the GPU of the same process (when run on GPU).

## 6 Conclusion

We introduced Bayesian compression, a way to tackle efficiency and compression in deep neural networks in a unified and principled way. Our proposed methods allow for theoretically principled compression of neural networks, improved energy efficiency with reduced computation while naturally learning the bit precisions for each weight. This serves as a strong argument in favor of Bayesian methods for neural networks, when we are concerned with compression and speed up.

**Acknowledgments**

We would like to thank Dmitry Molchanov, Dmitry Vetrov, Klamer Schutte and Dennis Koelma for valuable discussions and feedback. This research was supported by TNO, NWO and Google.

## Footnotes

*Canadian Institute For Advanced Research.

[2]In practice this term is a large constant determined by the weight precision.

[3]Stricly speaking the result of eq. 4 only holds when each weight has its own scale and not when that scale is shared across multiple weights. Nevertheless, in practice we obtain a prior that behaves in a similar way, i.e. it biases the variational posterior to be sparse.

[4]$\sigma(x) = (1 + \exp(-x))^{-1}$, $m(x) = \log(1 + \exp(x))$

[5]$\mathbb{V}(w_{ij}) = \mathbb{V}(z_i\tilde{w}_{ij}) = \mathbb{V}(z_i)\left(\mathbb{E}[\tilde{w}_{ij}]^2 + \mathbb{V}(\tilde{w}_{ij})\right) + \mathbb{V}(\tilde{w}_{ij})\mathbb{E}[z_i]^2$.

[6]Notice that the fact that we are using mean-field variational approximations (which we chose for simplicity) can potentially underestimate the variance, thus lead to higher bit precisions for the weights. We leave the exploration of more involved posteriors for future work.

[7]The product of log-normal r.v.s is another log-normal and a power of a log-normal r.v. is another log-normal.

[8]Empirically, it slightly better separates the scales compared to the negative log-mean $-(\mu_{z_i} + 0.5\sigma_{z_i}^2)$.

[9]`https://github.com/BVLC/caffe/tree/master/examples/mnist`

[10]The adapted CIFAR 10 version described at `http://torch.ch/blog/2015/07/30/cifar.html`.

[11]We also tried to finetune the same network with Sparse VD, but unfortunately it increased the error considerably (around 3% extra error), therefore we do not report those results.

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
