[Supplementary Material]

# Bayesian Compression for Deep Learning: Appendix

**Christos Louizos**
University of Amsterdam
TNO Intelligent Imaging
c.louizos@uva.nl

**Karen Ullrich**
University of Amsterdam
k.ullrich@uva.nl

**Max Welling**
University of Amsterdam
CIFAR*
m.welling@uva.nl

## A. Detailed experimental setup

We implemented our methods in Tensorflow [1] and optimized the variational parameters using Adam [5] with the default hyperparameters. The means of the conditional Gaussian $q_\phi(\mathbf{W}|\mathbf{z})$ were initialized with the scheme proposed at [4], whereas the log of the standard deviations were initialized by sampling from $\mathcal{N}(-9, 1e-4)$. The parameters of $q_\phi(\mathbf{z})$ were initialized such that the overall mean of $\mathbf{z}$ is $\approx 1$ and the overall variance is very low ($\approx 1e-8$); this ensures that all of the groups are active during the initial training iterations.

As for the standard deviation constraints; for the LeNet-300-100 architecture we constrained the standard deviation of the first layer to be $\leq 0.2$ whereas for the LeNet-5-Caffe we constrained the standard deviation of the first layer to be $\leq 0.5$. The remaining standard deviations were left unconstrained. For the VGG network we constrained the standard deviations of the 64 and 128 feature map layers to be $\leq 0.1$, the standard deviations of the 256 feature map layers to be $\leq 0.2$ and left the rest of the standard deviations unconstrained. We also found beneficial the incorporation of "warm-up" [8], i.e we annealed the negative KL-divergence from the prior to the approximate posterior with a linear schedule for the first 100 epochs. We initialized the means of the approximate posterior by the weights and biases obtained from a VGG network trained with batch normalization and dropout on CIFAR 10. For our method we disabled batch-normalization during training.

As for preprocessing the data; for MNIST the only preprocessing we did was to rescale the digits to lie at the $[-1, 1]$ range and for CIFAR 10 we used the preprocessed dataset provided by [9].

Furthermore, do note that by pruning a given filter at a particular convolutional layer we can also prune the parameters corresponding to that feature map for the next layer. This similarly holds for fully connected layers; if we drop a given input neuron then the weights corresponding to that node from the previous layer can also be pruned.

## B. Standards for Floating-Point Arithmetic

Floating points values eventually need to be represented in a binary basis in a computer. The most common standard today is the IEEE 754-2008 convention [7]. It defines $x$-bit base-2 formats, officially referred to as binary$x$, with $x \in \{16, 32, 64, 128\}$. The formats are also widely known as half, single, double and quadruple precision floats, respectively and used in almost all programming languages as a standard. The format considers 3 kinds of bits: one sign bit, $w$ exponent bits and $p$ precision bits.

The Sign bit determines the sign of the number to be represented. The exponent $E$ is an $w$-bit signed integer, e.g. for single precision $w = 8$ and thus $E \in [-127, 128]$. In practice, exponents range from is smaller since the first and the last number are reserved for special numbers. The true significand or mantissa includes t bits on the right of the binary point. There is an implicit leading bit with value

Figure 1: A symbolic representation of the binary$x$ format [7].

Table 1: Floating point formats

| Bits per Float | Exponent width [bit] | Significand precision [bit] | underflow level | overflow level | unit roundoff |
|---|---|---|---|---|---|
| 64 | 11 | 52 | $2.22 \times 10^{-308}$ | $1.79 \times 10^{308}$ | $2.22 \times 10^{-16}$ |
| 32 | 8 | 23 | $1.17 \times 10^{-38}$ | $3.40 \times 10^{38}$ | $1.19 \times 10^{-7}$ |
| 16 | 5 | 10 | $6.10 \times 10^{-05}$ | $6.54 \times 10^{4}$ | $9.76 \times 10^{-4}$ |

one. A values is consequently decomposed as follows

$$\text{mantissa} = 1 + \sum_{i=1}^{t} b_i 2^{-i} \tag{1}$$

$$\text{value} = (-1)^{\text{sign bit}} \times 2^{E} \times \text{mantissa}. \tag{2}$$

In table 1, we summarize common and less common floating point formats.

There is however the possibility to design a self defined format. There are 3 important quantities when choosing the right specification: overflow, underflow and unit round off also known as machine precision. Each one can be computed knowing the number of exponent and significant bits. in our work for example we consider a format that uses significantly less exponent bits since network parameters usually vary between [-10,10]. We set the unit round off equal to the precision and thus can compute the significant bits necessary to represent a specific weight.

Beyond designing a tailored floating point format for deep learning, recent work also explored the possibility of deep learning with mixed formats [6, 3]. For example, imagine the activations having high precision while weights can be low precision.

## C. Shrinkage properties of the normal-Jeffreys and horseshoe priors

(a) Empirical CDF

(b) Prior on shrinkage coefficient

Figure 2: Comparison of the behavior of the log-uniform / normal-Jeffreys (NJ) prior and the horseshoe (HS) prior (where $s = 1$). Both priors behave similarly at zero but the normal-Jeffreys has an extremely heavy tail (thus making it non-normalizable).

In this section we will provide some insights about the behavior of each of the priors we employ by following the excellent analysis of [2]; we can perform a change of variables and express the scale mixture distribution of eq.3 in the main paper in terms of a shrinkage coefficient, $\lambda = \frac{1}{1+z^2}$:

$$\lambda \sim p(\lambda); \qquad w \sim \mathcal{N}\left(0, \frac{1-\lambda}{\lambda}\right). \tag{3}$$

It is easy to observe that eq. 3 corresponds to a continuous relaxation of the spike-and-slab prior: when $\lambda = 0$ we have that $p(w|\lambda = 0) = \mathcal{U}(-\infty, \infty)$, i.e. no shrinkage/regularization for $w$, when $\lambda = 1$ we have that $p(w|\lambda = 1) = \delta(w = 0)$, i.e. $w$ is exactly zero, and when $\lambda = \frac{1}{2}$ we have that $p(w|\lambda = \frac{1}{2}) = \mathcal{N}(0, 1)$. Now by examining the implied prior on the shrinkage coefficient $\lambda$ for both the log-uniform and the horseshoe priors we can better study their behavior. As it is explained at [2], the half-Cauchy prior on $z$ corresponds to a beta prior on the shrinkage coefficient, $p(\lambda) = \mathcal{B}(\frac{1}{2}, \frac{1}{2})$, whereas the normal-Jeffreys / log-uniform prior on $z$ corresponds to $p(\lambda) = \mathcal{B}(\epsilon, \epsilon)$ with $\epsilon \approx 0$. The densities of both of these distributions can be seen at Figure 2b. As we can observe, the log-uniform prior posits a distribution that concentrates almost all of its mass at either $\lambda \approx 0$ or $\lambda \approx 1$, essentially either pruning the parameter or keeping it close to the maximum likelihood estimate due to $p(w|\lambda \approx 1) = \mathcal{U}(-\infty, \infty)$. In contrast the horseshoe prior maintains enough probability mass for the in-between values of $\lambda$ and thus can, potentially, offer better regularization and generalization.

**D. Negative KL-divergences for log-normal approximating posteriors**

Let $q(z) = \mathcal{LN}(\mu, \sigma^2)$ be a log-normal approximating posterior. Here we will derive the negative KL-divergences to $q(z)$ from inverse gamma, gamma and half-normal distributions.

Let $p(z)$ be an inverse gamma distribution, i.e. $p(z) = \mathcal{IG}(\alpha, \beta)$. The negative KL-divergence can be expressed as follows:

$$-KL(q(z)||p(z)) = \int q(z) \log p(z)dz - \int q(z) \log q(z)dz. \tag{4}$$

The second term is the entropy of the log-normal distribution which has the following form:

$$\mathcal{H}_q = -\int q(z) \log q(z)dz = \frac{1}{2} \log \sigma^2 + \mu + \frac{1}{2} + \frac{1}{2} \log(2\pi). \tag{5}$$

The first term is the negative cross-entropy of the log-normal approximate posterior from the inverse-Gamma prior:

$$-\mathcal{CE}_{qp} = \int q(z) \left( \alpha \log \beta - \log \Gamma(\alpha) - (\alpha + 1) \log z - \frac{\beta}{z} \right) dz \tag{6}$$

$$= \alpha \log \beta - \log \Gamma(\alpha) - (\alpha + 1) \mathbb{E}_{q(z)}[\log z] - \beta \mathbb{E}_{q(z)}[z^{-1}]. \tag{7}$$

Since the natural logarithm of a log-normal distribution $\mathcal{LN}(\mu, \sigma^2)$ follows a normal distribution $\mathcal{N}(\mu, \sigma^2)$ we have that $\mathbb{E}_{q(z)}[\log z] = \mu$. Furthermore we have that if $x \sim \mathcal{LN}(\mu, \sigma^2)$ then $\frac{1}{x} \sim \mathcal{LN}(-\mu, \sigma^2)$, therefore $\mathbb{E}_{q(z)}[z^{-1}] = \exp(-\mu + \frac{\sigma^2}{2})$. Putting everything together we have that:

$$-\mathcal{CE}_{qp} = \alpha \log \beta - \log \Gamma(\alpha) - (\alpha + 1)\mu - \beta \exp(-\mu + \frac{\sigma^2}{2}). \tag{8}$$

Therefore the negative KL-divergence is:

$$-KL(q(z)||p(z)) = \alpha \log \beta - \log \Gamma(\alpha) - \alpha\mu - \beta \exp(-\mu + 0.5\sigma^2) +$$
$$+ 0.5(\log \sigma^2 + 1 + \log(2\pi)). \tag{9}$$

Now let $p(z)$ be a Gamma prior, i.e. $p(z) = \mathcal{G}(\alpha, \beta)$. We have that the negative cross-entropy changes to:

$$-\mathcal{CE}_{qp} = \int q(z) \left( -\alpha \log \beta - \log \Gamma(\alpha) - \frac{z}{\beta} + (\alpha - 1) \log z \right) dz \tag{10}$$

$$= -\alpha \log \beta - \log \Gamma(\alpha) - \beta^{-1} \mathbb{E}_{q(z)}[z] + (\alpha - 1) \mathbb{E}_{q(z)}[\log z] \tag{11}$$

$$= -\alpha \log \beta - \log \Gamma(\alpha) - \beta^{-1} \exp(\mu + \frac{\sigma^2}{2}) + (\alpha - 1)\mu. \tag{12}$$

Therefore the negative KL-divergence is:

$$-KL(q(z)||p(z)) = -\alpha \log \beta - \log \Gamma(\alpha) + \alpha\mu - \beta^{-1} \exp(\mu + 0.5\sigma^2) +$$
$$+ 0.5(\log \sigma^2 + 1 + \log(2\pi)). \tag{13}$$

Now, by employing the aforementioned we can express the negative KL-divergence from $p(s_a, s_b, \tilde{\alpha}, \tilde{\beta})$ to $q_\phi(s_a, s_b, \tilde{\alpha}, \tilde{\beta})$ as follows:

$$-KL(q_\phi(s_a)||p(s_a)) = \log \tau_0 - \tau_0^{-1} \exp\left(\mu_{s_a} + \frac{1}{2}\sigma_{s_a}^2\right) + \frac{1}{2}\left(\mu_{s_a} + \log \sigma_{s_a}^2 + 1 + \log 2\right) \quad (14)$$

$$-KL(q_\phi(s_b)||p(s_b)) = -\exp\left(\frac{1}{2}\sigma_{s_b}^2 - \mu_{s_b}\right) + \frac{1}{2}\left(-\mu_{s_b} + \log \sigma_{s_b}^2 + 1 + \log 2\right) \quad (15)$$

$$-KL(q_\phi(\tilde{\alpha})||p(\tilde{\alpha})) = \sum_i^A \left(-\exp\left(\mu_{\tilde{\alpha}_i} + \frac{1}{2}\sigma_{\tilde{\alpha}_i}^2\right) + \frac{1}{2}\left(\mu_{\tilde{\alpha}_i} + \log \sigma_{\tilde{\alpha}_i}^2 + 1 + \log 2\right)\right) \quad (16)$$

$$-KL(q_\phi(\tilde{\beta})||p(\tilde{\beta})) = \sum_i^A \left(-\exp\left(\frac{1}{2}\sigma_{\tilde{\beta}_i}^2 - \mu_{\tilde{\beta}_i}\right) + \frac{1}{2}\left(-\mu_{\tilde{\beta}_i} + \log \sigma_{\tilde{\beta}_i}^2 + 1 + \log 2\right)\right), \quad (17)$$

with the KL-divergence for the weight distribution $q_\phi(\tilde{\mathbf{W}})$ given by eq.8 in the main paper.

## E. Visualizations

Figure 3: Distribution of the thresholds for the Sparse Variational Dropout 3a, Bayesian Compression with group normal-Jeffreys (BC-GNJ) 3b and group Horseshoe (BC-GHS) 3c priors for the three layer LeNet-300-100 architecture. It is easily observed that there are usually two well separable groups with BC-GNJ and BC-GHS, thus making the choice for the threshold easy. Smaller values indicate signal whereas larger values indicate noise (i.e. useless groups).

Figure 4: Distribution of the bit precisions for the Sparse Variational Dropout 4a, Bayesian Compression with group normal-Jeffreys (BC-GNJ) 4b and group Horseshoe (BC-GHS) 4c priors for the three layer LeNet-300-100 architecture. All of the methods usually require far fewer than 32bits for the weights.

## F. Algorithms for the feedforward pass

Algorithms 1, 2, 3, 4 describe the forward pass using local reparametrizations for fully connected and convolutional layers with the approximate posteriors for the Bayesian Compression (BC) with group normal-Jeffreys (BC-GNJ) and group Horseshoe (BC-GHS) priors employed at the experiments. For the fully connected layers we coupled the scales for each input neuron whereas for the convolutional we couple the scales for each output feature map. $\mathbf{M}_w, \mathbf{\Sigma}_w$ are the means and variances of each layer, $\mathbf{H}$ is a minibatch of activations of size $K$. For the first layer we have that $\mathbf{H} = \mathbf{X}$ where $\mathbf{X}$ is the minibatch of inputs. For the convolutional layers $N_f$ are the number of convolutional filters, $*$ is the convolution operator and we assume the [batch, height, width, feature maps] convention.

| **Algorithm 1** Fully connected BC-GNJ layer $h$. | **Algorithm 2** Convolutional BC-GNJ layer $h$. |
|---|---|
| **Require:** $\mathbf{H}, \mathbf{M}_w, \boldsymbol{\Sigma}_w$ | **Require:** $\mathbf{H}, \mathbf{M}_w, \boldsymbol{\Sigma}_w$ |
| 1: $\hat{\mathbf{E}} \sim \mathcal{N}(0,1)$ | 1: $\mathbf{M}_h = \mathbf{H} * \mathbf{M}_w$ |
| 2: $\mathbf{Z} = \boldsymbol{\mu}_z + \boldsymbol{\sigma}_z \odot \hat{\mathbf{E}}$ | 2: $\mathbf{V}_h = \mathbf{H}^2 * \boldsymbol{\Sigma}_w$ |
| 3: $\hat{\mathbf{H}} = \mathbf{H} \odot \mathbf{Z}$ | 3: $\hat{\mathbf{E}} \sim \mathcal{N}(0,1)$ |
| 4: $\mathbf{M}_h = \hat{\mathbf{H}} \mathbf{M}_w$ | 4: $\hat{\boldsymbol{\mu}}_z = \text{reshape}(\boldsymbol{\mu}_z, [K,1,1,N_f])$ |
| 5: $\mathbf{V}_h = \hat{\mathbf{H}}^2 \boldsymbol{\Sigma}_w$ | 5: $\hat{\boldsymbol{\sigma}}_z = \text{reshape}(\boldsymbol{\sigma}_z, [K,1,1,N_f])$ |
| 6: $\mathbf{E} \sim \mathcal{N}(0,1)$ | 6: $\mathbf{Z} = \hat{\boldsymbol{\mu}}_z + \hat{\boldsymbol{\sigma}}_z \odot \hat{\mathbf{E}}$ |
| 7: return $\mathbf{M}_h + \sqrt{\mathbf{V}_h} \odot \mathbf{E}$ | 7: $\mathbf{E} \sim \mathcal{N}(0,1)$ |
|  | 8: return $\mathbf{M}_h \odot \mathbf{Z} + \sqrt{\mathbf{V}_h \odot \mathbf{Z}^2} \odot \mathbf{E}$ |

| **Algorithm 3** Fully connected BC-GHS layer $h$. | **Algorithm 4** Convolutional BC-GHS layer $h$. |
|---|---|
| **Require:** $\mathbf{H}, \mathbf{M}_w, \boldsymbol{\Sigma}_w$ | **Require:** $\mathbf{H}, \mathbf{M}_w, \boldsymbol{\Sigma}_w$ |
| 1: $\hat{\boldsymbol{\epsilon}} \sim \mathcal{N}(0,1)$ | 1: $\mathbf{M}_h = \mathbf{H} * \mathbf{M}_w$ |
| 2: $\mu_s = .5\mu_{s_a} + .5\mu_{s_b}$ | 2: $\mathbf{V}_h = \mathbf{H}^2 * \boldsymbol{\Sigma}_w$ |
| 3: $\sigma_s = \sqrt{.25\sigma_{s_a}^2 + .25\sigma_{s_b}^2}$ | 3: $\hat{\boldsymbol{\epsilon}} \sim \mathcal{N}(0,1)$ |
| 4: $\log \mathbf{s} = \mu_s + \sigma_s \odot \hat{\boldsymbol{\epsilon}}$ | 4: $\mu_s = .5\mu_{s_a} + .5\mu_{s_b}$ |
| 5: $\boldsymbol{\mu}_{\tilde{z}} = .5\boldsymbol{\mu}_{\tilde{\alpha}} + .5\boldsymbol{\mu}_{\tilde{\beta}} + \log \mathbf{s}$ | 5: $\sigma_s = \sqrt{.25\sigma_{s_a}^2 + .25\sigma_{s_b}^2}$ |
| 6: $\boldsymbol{\sigma}_{\tilde{z}} = \sqrt{.25\boldsymbol{\sigma}_{\tilde{\alpha}}^2 + .25\boldsymbol{\sigma}_{\tilde{\beta}}^2}$ | 6: $\log \mathbf{s} = \text{reshape}(\mu_s + \sigma_s \odot \hat{\boldsymbol{\epsilon}}, [K,1,1,1])$ |
| 7: $\hat{\mathbf{E}} \sim \mathcal{N}(0,1)$ | 7: $\boldsymbol{\mu}_{\tilde{z}} = \text{reshape}(.5\boldsymbol{\mu}_{\tilde{\alpha}} + .5\boldsymbol{\mu}_{\tilde{\beta}}, [K,1,1,N_f])$ |
| 8: $\mathbf{Z} = \exp(\boldsymbol{\mu}_{\tilde{z}} + \boldsymbol{\sigma}_{\tilde{z}} \odot \hat{\mathbf{E}})$ | 8: $\boldsymbol{\sigma}_{\tilde{z}} = \text{reshape}(\sqrt{.25\boldsymbol{\sigma}_{\tilde{\alpha}}^2 + .25\boldsymbol{\sigma}_{\tilde{\beta}}^2}, [K,1,1,N_f])$ |
| 9: $\hat{\mathbf{H}} = \mathbf{H} \odot \mathbf{Z}$ | 9: $\hat{\mathbf{E}} \sim \mathcal{N}(0,1)$ |
| 10: $\mathbf{M}_h = \hat{\mathbf{H}} \mathbf{M}_w$ | 10: $\mathbf{Z} = \exp(\boldsymbol{\mu}_{\tilde{z}} + \log \mathbf{s} + \boldsymbol{\sigma}_{\tilde{z}} \odot \hat{\mathbf{E}})$ |
| 11: $\mathbf{V}_h = \hat{\mathbf{H}}^2 \boldsymbol{\Sigma}_w$ | 11: $\mathbf{E} \sim \mathcal{N}(0,1)$ |
| 12: $\mathbf{E} \sim \mathcal{N}(0,1)$ | 12: return $\mathbf{M}_h \odot \mathbf{Z} + \sqrt{\mathbf{V}_h \odot \mathbf{Z}^2} \odot \mathbf{E}$ |
| 13: return $\mathbf{M}_h + \sqrt{\mathbf{V}_h} \odot \mathbf{E}$ |  |

## Footnotes

*Canadian Institute For Advanced Research.