[Reviews · NeurIPS 2017]

Reviewer 1



This paper approaches model compression using a group sparsity prior, to allow entire columns rather than just individual weights to be dropped out. They also use the variance of the posterior distribution over weights to automatically set the precision for fixed point weight quantization. The underlying ideas seem good, and the experimental results seem promising. However, the paper supports the core idea with a great deal of mathematical complexity. The math was presented in a way that I often found confusing, and in several places seems either wrong or poorly motivated (e.g., KL divergences are negative, right and left side of equations are not equal, primary motivation for model compression given in terms of minimum description length). (the paper would have been much stronger had it stuck with just the core idea and supporting experimental results!) Section 2 – This seems like a strange primary motivation for model compression. Our goal when using deep networks is (almost) never to transmit a set of conditional labels with the fewest bits, and almost always to accurately assign new labels to new data points. 115 - its’ -> its 128/129 - I didn’t understand the sentence. Probably you mean to say that the variance of the spike rather than the variance of the slab is zero? 138 - This is not a fully Bayesian treatment. The posterior is being restricted to a specific approximate hierarchical functional form. Eq 4 - We only recover Jeffrey’s prior in the 1D case. We do not recover Jeffrey’s prior e.g. if we compute p(W) from equation 5. 154 - I don’t understand the correspondence to dropout here. I believe that would require that the distribution included a delta function at zero variance. Eq 9. - If alpha_i =1, the KL divergence is negative, and as alpha_i --> 0, the KL goes to -infinity. KL should be strictly nonnegative, so something is definitely wrong here:. Eq 10 - The middle and the rightmost terms are not equal to each other. The middle term sets all off diagonal weights to zero. (Should this be ordinary matrix multiplication) 186 – S has a heavy tail. This will encourage *all* weights to be simultaneously large occasionally. 197 – what does it mean for the shape hyperprior to go to zero? Eq 19 - Middle term has the same problem as in equation 10. (239 - aside, largely tangential to current paper -- it's strange how 5 bits keeps on coming up as a per-weight capacity limit, in a variety of architectures and settings. It seems suggestive of something more fundamental going on. e.g. [Han et al, 2015], [Bartol et al, 2016], [Collins, Sohl-Dickstein, Sussillo, 2016])

Reviewer 2



This paper describes a method for compressing a neural network: pruning its neurons and reducing the floating point precision of its weights to make it smaller and faster at test time. The method works by training the neural net with SVI using a factored variational approximation that shares a scale parameter between sets of weights. The entropy term of the variational objective incentivizes weights to have a large variance, which leads to compression by two separate mechanisms: neurons with high-variance weights relative to their means can be pruned, and layers can have their floating point precision reduced to a level commensurate with the variance of their weights. It's a fairly principled approach in which compression arises naturally as a consequence of Bayesian uncertainty. It also seems effective in practice. I have three main reasons for not giving a higher score. First, I'm not convinced that the work is hugely original. It leans very heavily on previous work, particularly Kingma et al. on variational dropout and Molchanov et al. on sparsification arising from variational dropout. The two novelties are (1) hierarchical priors and (2) reduced floating point precision, which are certainly useful but they are unsurprising extensions of the previous work. Additionally, the paper does a poor job of placing itself in the context of these previous papers. Despite repeatedly referencing the variational dropout papers for forms of variational approximations or derivations, the concept of 'variational dropout' is never actually introduced! The paper would be much clearer if it at least offered a few-sentence summary of each of these previous papers, and described where it departs from them. Second, the technical language is quite sloppy in places. The authors regularly use the phrase "The KL divergence between X and Y" instead of "the KL divergence from X to Y". "Between" just shouldn't be used because it implies symmetry. Even worse, in all cases, the order of "X" and "Y" is backwards, so that even "from X to Y" would be incorrect. Another example of sloppy language is the use of the term "posterior" to describe the variational approximation q. This causes genuine confusion, because the actual posterior is a very relevant object. At the very least, the term should be "posterior approximation". In the paper's defense, the actual equations are generally clear and unambiguous. Finally, the experiments are a bit unimaginative. For example, although the authors argue that Bayesian inference and model compression are well aligned, we know they have quite different goals and you would expect to find some situations where they could be in conflict. Why not explore this tradeoff by looking at performance and compression as a function of hyperparameters like $\tau_0$? How much compression do you get if you select $\tau_0$ by maximizing the marginal likelihood estimate rather than choosing an arbitrary value?

Reviewer 3



1) Despite the abstract putting equal weight on pruning and precision reduction, the latter seems to have been relegated to the experiments section like a bit of an afterthought. I understand there are some implementation difficulties, but if it's part of the main thesis it should feature more prominently. Also, the description of how to handle that is very superficial. You refer to the appendix, but I'd generally say that central components should be kept in the main text. (although I can sympathize with the space constraints) 2) You're using variational inference, so I would have liked to see a few words about how mean-field assumptions and variance underestimation (your q is unlikely to be flexible enough to avoid this problem) affect you given that you use variance so explicitly. Especially bit precision inference seems like it could be sensitive to this? 3) what are your justifications for the choice of q approximations? In particular their form and component distributions. Overall, I think it was an excellent paper, if a bit dense with content.